# Voltage Sensing in Bacterial Protein Translocation

**DOI:** 10.3390/biom10010078

**Published:** 2020-01-03

**Authors:** Denis G. Knyazev, Roland Kuttner, Ana-Nicoleta Bondar, Mirjam Zimmerman, Christine Siligan, Peter Pohl

**Affiliations:** 1Institute of Biophysics, Johannes Kepler University Linz, Gruberstr. 40, 4020 Linz, Austria; 2Theoretical Molecular Biophysics Group, Department of Physics, Freie Universität Berlin, Arnimallee 14, D-14195 Berlin, Germany

**Keywords:** translocon, SecY, Sec61, gating

## Abstract

The bacterial channel SecYEG efficiently translocates both hydrophobic and hydrophilic proteins across the plasma membrane. Translocating polypeptide chains may dislodge the plug, a half helix that blocks the permeation of small molecules, from its position in the middle of the aqueous translocation channel. Instead of the plug, six isoleucines in the middle of the membrane supposedly seal the channel, by forming a gasket around the translocating polypeptide. However, this hypothesis does not explain how the tightness of the gasket may depend on membrane potential. Here, we demonstrate voltage-dependent closings of the purified and reconstituted channel in the presence of ligands, suggesting that voltage sensitivity may be conferred by motor protein SecA, ribosomes, signal peptides, and/or translocating peptides. Yet, the presence of a voltage sensor intrinsic to SecYEG was indicated by voltage driven closure of pores that were forced-open either by crosslinking the plug to SecE or by plug deletion. We tested the involvement of SecY’s half-helix 2b (TM2b) in voltage sensing, since clearly identifiable gating charges are missing. The mutation L80D accelerated voltage driven closings by reversing TM2b’s dipolar orientation. In contrast, the L80K mutation decelerated voltage induced closings by increasing TM2b’s dipole moment. The observations suggest that TM2b is part of a larger voltage sensor. By partly aligning the combined dipole of this sensor with the orientation of the membrane-spanning electric field, voltage may drive channel closure.

## 1. Introduction

Subsequent to their synthesis, many of the bacterial membrane and secretory proteins take the translocation pathway across the heterotrimeric SecYEG complex of the evolutionary conserved Sec family [1,2]. The bacterial SecYEG has a striking homology to its archaeal and eukaryotic protein family members SecYEβ and Sec61αβγ. The main translocation unit of the translocon, SecY, has a central pore, which is closed for the passage of small molecules in its resting state by a hydrophobic ring (HR) of six amino acids and by a re-entrant loop (TM2a), also called the plug domain (PD) [3]. SecY can be seen as a clamshell with a hinge clamped by SecE, and a lateral gate with helixes 2b and 7 acting as gate posts [4]. The channel opens upon binding of the signal sequence of translocating peptides between the gate posts [5,6]. This enables subsequent polypeptide segments to translocate through SecY’s central pore [7]. Hydrophobic helices exit via the lateral gate into the lipid phase, while hydrophilic segments enter the periplasm [8].

Structural investigations suggested the HR be important for maintaining the membrane barrier to small molecules during membrane translocation of much larger polypeptide chains [8,9]. This is in line with functional experiments on planar bilayers and molecular dynamics simulations, in which the substitution of pore ring isoleucines for more hydrophilic residues rendered the purified and reconstituted SecYEG complex leaky to small molecules [3,6,10]. The same observation was made with the active translocon in spheroplasts [11].

Yet, the activated wild type SecYEG complex also appeared to be leaky, provided the absolute values of membrane potential *φ* dropped below 100 mV [6,12]. The substitution of hydrophobic by hydrophilic residues in the HR only served to increase the leak [6]. Interestingly, the idle SecYEG complex with ribosomes [13] or signal peptides [6] also showed voltage-dependent ion channel activity. In contrast, we found no voltage-dependence of the wild-type SecYEG complex in the absence of a ligand, i.e., the channel cannot be opened by voltage [3]. These observations agree well with the lack of obvious gating charges, and suggest that the actual voltage sensor may not be part of the SecYEG complex, but may originate from the ribosome or the signal peptide.

Alternatively, the sensor may be part of the SecYEG complex. If so, the voltage sensor should be located in the transmembrane region to sense *φ*. This conclusion rules out SecG’s cytoplasmic loop, which was suggested to be capable of sealing SecY during translocation [14]. Theoretically, the PD could act as the voltage sensor. It contains a conserved arginine at position 74 that, if moved by the electric field, would be able to reversibly close and open the channel. We tested this hypothesis by restricting PD’s mobility or by deleting the PD altogether. 

Our experiments with purified and reconstituted SecYEG complexes and their ligands suggested a rather uncommon mechanism of voltage sensing: voltage acts to align protein dipoles with the transmembrane electric field, thereby reverting the movement they undergo during channel opening.

## 2. Materials and Methods 

### 2.1. SecYEG Purification

The purification of SecYEG variants (wild type, SecY(S329C)EG, SecY(Δ60-74)EG, SecY(L80K)EG, SecY(L80D)EG, SecY(F67C)E(S120C)G) was performed as previously described [6,13]. Point mutations in SecYEG where introduced using PCR mediated site directed mutagenesis and verified by sequencing. Briefly SecYEG was overexpressed for 3 h in *E. coli* c43 (DE3) cells from a pBad22 vector and induced with 2 g/L of arabinose. The collected cells were lysed with an Emulsiflex homogenizer (Avestin, Ottawa, Canada) in 20 mM Tris (pH 7,5); 300 mM NaCl; 10% glycerol; supplemented with complete protease inhibitor (Roche, Basel, Switzerland). The membrane fraction was pelleted at 100,000× *g* and solvated in 1% (w/v) Dodecyl-malto-pyranoside (DDM, Anatrace, Maumee, OH, USA). Affinity chromatography with Ni-NTA-Agarose (Quiagen, Hilden, Germany) and size exclusion chromatography were used to improve sample purity. SecY(S329C)EG and SecY(F67C)E(S120C)G were purified and reconstituted in the presence of 0.4 mM TCEP to ensure a reduced state.

### 2.2. SecYEG Reconstitution into Lipid Vesicles

SecYEG was reconstituted into *E. coli* polar lipid extract (Avanti Polar Lipids, Alabaster, AL, USA) vesicles pre-dissolved in deoxy-BigChap (Anatrace, Maumee, OH, USA) as previously described [3]. Biobeads SM2 (Biorad, Hercules, CA, USA) were added to remove the excess detergent and the resulting turbid suspension was pelleted at 100.000 g. The resulting pellet was resuspended and extruded through a 100 nm filter. Mass ratios of protein to lipid of 1:54 to 1:108 were used.

### 2.3. ProOmpA-DHFR (pOD) Purification

Subsequently, (pOD) was purified as previously described [6]. The construct is composed up of the first 69 amino acids of OmpA followed by full length dihydrofolate reductase (DHFR) and a 6x His tag for affinity purification in a pBad22 vector backbone. DHFR improves the water solubility of the hybrid. Moreover, it cannot be unfolded when bound to Methotrexate and thus blocks translocation. MM52 cells transformed with the target plasmid were grown at 30 °C in 2xYT Ampicilin (100 mg/L) medium till they reached an OD600 of about 1. Subsequently the suspension was diluted tenfold with fresh medium and incubated for 30′ at 37 °C, followed by overexpression for 2 h induced with 2 g/L of arabinose. Cells were lysed by homogenization in 50 mM Tris (pH 7.5), 300 mM KCl, 10% glycerol, 1 mM TCEP and protease inhibitor. After non soluble cell components were removed by centrifugation (100,000× *g*, 1 h), Ni-NTA agarose (Quiagen, Hilden, Germany) was used for affinity chromatography. Finally, size exclusion chromatography (Äkta, GE Healthcare, Chicago, IL, USA) was exploited to improve purity.

### 2.4. SecA Purification

We followed our protocol for the purification of both SecA [6] with minor modifications: SecA was expressed in Nico21 (DE3) cells allowing for removal of impurities such as metal binding proteins, normally co-purified during Ni-NTA affinity chromatography, with chitin beads (New England Biolabs). This was followed by size exclusion chromatography to further increase purity. 

### 2.5. Translocation Assay

We monitored translocation of pOD fusion proteins into reconstituted lipid vesicles to ensure SecYEG’s functionality as previously described [15,16]. The uptake experiments were performed in the presence of SecA and ATP (Appendix A). After an incubation period, pOD molecules that did not translocate into the vesicles were digested by proteinase K.

### 2.6. RNC Purification

We purified FtsQ-RNC constructs as previously described [12]. These RNCs contained nascent chain based on 101 N-terminal residues of FtsQ. This RNC contained in addition to the His tag on the ribosome a calmodulin binding peptide for an additional affinity purification step. The specific stalling sequence motif SecM contained the FXXXXWIXXXXGIRAGP motif. Nascent chains were expressed for 2 h in JE28 cells containing his-tagged ribosomes [17] by addition of 2 g/L arabinose. Pelleted cells were lysed in 20 mM Tris (pH 7.6), 10 mM MgCl2, 150 mM KCl, and 30 mM NH4Cl in a homogenizer at low pressure (15–17 kPsi). Two subsequent affinity purifications with Ni-NTAagarose (Quiagen, Hilden, Germany) and calmodulin agarose (Sigma-Aldrich, St. Louis, MO, USA) served to first selectively purify ribosomes and second to select those ribosomes that contained a stalled nascent chain.

### 2.7. Signal Peptide

For cross-linking a signal peptide to SecY(S329C)EG, we used the signal peptide from the precursor form of outer membrane protein A (proOmpA-SP). This contained a cysteine instead of an alanine in the last position: MKKTAIAIAVALAGFATVAQC (synthesized by Peptide 2.0 Inc., Chantilly, VA, USA). Disulfide bridge formation was induced by potassium tetrathionate (KTT).

### 2.8. Purification of Empty Ribosomes

Bacterial ribosomes were purified from E. coli MRE600 as described previously [12]. The ribosomes contained a 6xHis-tag at the L7 protein of the 50S subunit for affinity chromatography. Cooling the cells to 4 °C during cell lysis terminated the translation. Incubation with 20 µM puromycin ensured the release of nascent chains from ribosomes.

### 2.9. Reconstitution of the SecYEG Complex into Planar Bilayers

“Solvent-free” planar bilayers were formed by combining two lipid monolayers from E.coli Polar lipid extract (Avanti Polar Lipids, Alabaster, AL, USA) in the aperture of a Teflon septum that separated two aqueous solutions from each other [18]. SecYEG containing proteoliposomes were fused to the free-standing planar bilayer as previously described [13]. The fusion was facilitated by osmotic pressure. The hyperosmotic compartment contained 450 mM KCl, 50 mM K-HEPES, proteoliposomes, 5 mM MgCl_2_, and the substrate (either 10 nM RNC, or 100 nM signal peptide, or 500 nM empty ribosomes, or 100 nM proOmpA-DHFR). The hypoosmotic compartment contained 150 mM KCl and 50 mM K-HEPES. Both compartments were kept at pH 7.5.

### 2.10. Electrophysiological Measurements

Ag/AgCl reference electrodes were immersed into the buffer solutions at both sides of the planar bilayers. The command electrode of the patch clamp amplifier (model EPC9, HEKA electronics, Chester, Canada) was dipped into the cis compartment and the ground electrode into the trans compartment. The recording filter for the transmembrane current was a 4-pole Bessel with −3 dB corner frequency of 0.1 kHz [19]. The raw data were analyzed using the TAC software package (Bruxton Corporation, Seattle, WA, USA). Gaussian filters of 12 Hz were applied to reduce noise. The data were then further processed using SigmaPlot (Systat Software Inc., San Jose, CA, USA).

### 2.11. Computation of Dipole Moments

Dipole moments were computed for TM2b of the *E. coli* SecY translocon. Close inspection of the only high-resolution structure PDB ID:5GAE indicated that the distance between the Cα atoms of P398 and F399 is ~8Å, which is too long for two amino acid residues that ought to be covalently bound to each other. We thus prepared a homology model of the *E. coli* SecY translocon using the sequence UniProt ID P0AGA2 as a template, and Phyre2 [20] for homology modeling. Hydrogen atoms were constructed using Chemistry at Harvard Molecular Mechanics (CHARMM) [21]. The protein was oriented along the membrane normal using visual molecular dynamics (VMD) [22] and placed with its center of mass in the center of coordinates. The L80D and L80K mutants of *E. coli* SecY were prepared using CHARMM. Partial atomic charges for the translocon atoms were taken from the CHARMM 36 all-atom force field [23,24,25]. The dipole moment computations were performed using VMD [22].

## 3. Results

### 3.1. Voltage Sensitivity of SecYEG in Complex with FtsQ-RNC

The addition of SecYEG containing proteoliposomes to planar bilayers did not elicit any channel activity (Figure 1A). This observation suggests that vesicle fusion to the planar bilayer did not occur because the SecYEG channel was closed and thus impermeable to the osmolyte. In contrast, we observed channel activity in the presence of RNC complexes (Figure 1B). That is, RNC binding to SecYEG triggered the opening of translocation channels, thereby allowing KCl to enter the vesicles. Subsequent vesicle swelling is known to facilitate vesicle fusion to planar bilayers [26]. The SecYEG-RNC channels showed infrequent closures at φ = −85 mV (Figure 1B). Their amplitude was similar to that of the SecYEG-preprotein or SecYEG-ribosome complexes [6,13]. Physiological φ values of −140 mV [27] decreased channel activity in a step-wise manner. Each closing matched the single channel amplitude (Figure 1C). The lifetime histogram revealed a decrease in channel lifetime to about 0.5 s (Figure 1D).

### 3.2. Voltage Sensitivity of SecYEG Is Not Granted by the Translocated Substrate

Voltage-dependent dislocation of the signal peptide from the lateral gate may be required for channel closure to occur. We tested the hypothesis by cross-linking the signal peptide of proOmpA with the lateral gate of SecY. To enable disulfide bridge formation, we first added one cysteine at position 21 to a peptide that consisted of the 20 N-terminal residues of proOmpA. Second we positioned the pairing cysteine in helix 8 of the translocon: SecY(S329C)EG. As described above, we overexpressed, purified and reconstituted the mutant protein. The proteoliposomes did not fuse to the planar bilayer in the absence of the signal peptide indicating that the mutant channel is closed in its resting state. Fusion occurred only in the presence of the signal peptide, as was evident from the observed channel activity (Figure 2). This observation is in line with the previously reported ability of signal peptides to open the translocon [5]. The SecYEG–signal peptide complex exhibited voltage sensitivity, i.e. channel activity ceased at higher *φ* values [6].

Cross-linking the cysteines of signal peptide and translocon by KTT did not abolish voltage sensitivity (Figure 2A). However, the characteristic lifetime of SecYEG channels significantly increased (Appendix A). It amounted to about 14 s, suggesting that the signal peptide in the lateral gate stabilizes the open conformation of SecYEG. Successful cross-linking of the signal peptide to the translocon was supported by the observation that all channels reopened, once *φ* was no longer applied (Figure 2C). In the absence of the cross-link only about two thirds of channels reopened [6], since the signal peptide was able to leave the lateral gate. Our crosslinking experiments indicate that (i) signal peptide release is not required for channel closure to occur and (ii) albeit charged, the signal peptide does not confer voltage sensitivity.

To confirm the hypothesis that neither the signal peptide nor the following polypeptide chain is essential for voltage sensing, we repeated the experiments with empty ribosomes. They were added to the proteoliposome-containing hypertonic compartment. Upon ribosome binding, the translocon opened and thus allowed the osmolyte to enter the vesicles. As a result, osmotically driven vesicle fusion to the planar bilayers occurred and we observed channel activity. Upon the application of φ = −120 mV, the channel closed with kinetics similar to those observed with RNCs (Figure 2D) indicating that voltage sensitivity is intrinsic to SecYEG. The experimental results are consistent with previously observed voltage sensitive closings of ribosome bound SecYEG [13]. The previous and the current sets of experiments differ in the sign of the applied potential indicating that the actual orientation of the electric field is of no importance (Appendix A). This observation indicates that SecYEG’s voltage sensor is very different from the voltage sensors of conventional ion channels. If added to the previously mentioned lack of obvious gating charges, this suggests that the voltage sensor may consist of electrical dipoles that orient themselves along the lines of the electric field.

### 3.3. PD Affects the Voltage Sensitivity of SecYEG

We tested whether the PD may act as voltage sensor. If the hypothesis was correct, the crosslink between the PD and SecE in the double cysteine mutant SecY(F67C)E(S120C)G [3] should result in an open voltage insensitive channel. Accordingly, we overexpressed and purified the mutant protein from E. coli and reconstituted it into lipid vesicles. When adding these vesicles in the presence of KTT to preformed planar lipid bilayers, we observed a stepwise increase in bilayer conductivity, indicating the fusion of vesicles containing the open channel (Figure 3A). The channels stayed permanently open at φ = −35 mV (Figure 3B). In contrast, we observed step-wise channel closings at φ = −140 mV (Figure 3C). Yet, channel lifetime amounted to several seconds and thus exceeded that measured for the wild-type SecYEG-RNC complex about tenfold (Figure 1C), suggesting (i) a physical connection between the PD and the actual voltage sensor or (ii) the PD being part of the voltage sensor. Moreover, channel lifetime displayed a much wider distribution (Figure 3D).

To rule out the remaining likelihood that SecYEG’s remarkable plasticity enables the plug to act as voltage sensor even when cross-linked to SecE, we overexpressed, purified and reconstituted a PD deficient mutant (SecY(Δ60–74)EG) [3]. An osmotic gradient served to fuse the proteoliposomes to the planar bilayer in the presence of proOmpA-DHFR, methotrexate, ATP, and SecA. Methotrexate binds to the polypeptide and thus prevents it from unfolding. As a result, we obtained a true translocation intermediate [6]. Nevertheless, we still observed voltage driven channel closings, albeit at a somehow reduced rate (Figure 3E), indicating that the PD is either part of the sensor or due to a physical connection to the sensor, which may slow down its movement.

### 3.4. Lateral Gate Helix 2b Is Part of the Voltage Sensor

Decelerated closings with the cross-linked PD pointed to TM2b as a likely candidate for the voltage sensor. The hypothesis is supported by the observation that TM2b adopts different positions in SecYEG’s open and closed states (Figure 4).

Since helix 2b is bare of charges, it may contribute to voltage sensitivity only by virtue of its dipole moment. To test the hypothesis, we substituted its N-terminal residue L80 by either lysine or aspartic acid. Purification from overexpressing *E. coli* cells and subsequent reconstitution of the mutant translocons into lipid vesicles was followed by fusion of these vesicles to planar bilayers. Electrophysiological recordings revealed accelerated closing kinetics of the L80D translocon (Figure 5A,B). In contrast, the L80K translocon exhibited decelerated closing kinetics (Figure 5C,D).

## 4. Discussion

The plasma membrane represents a barrier to ion movement [31]. SecYEG maintains this barrier only in energized membranes, i.e., when *φ* adopts large absolute values [6]. Below a threshold value of *φ*, a conformational transition occurs that transforms the translocon into an ion conducting channel. This is true only for channels engaged in polypeptide translocation or for channels that are bound to ribosomes, signal peptides or the motor protein SecA. In contrast, the ligand-free SecYEG blocks ion permeation independent of voltage.

We established that the voltage sensor is an intrinsic part of SecYEG. The conclusion is supported by the observation that the ligands are interchangeable—ribosomes, SecA, signal peptides and translocation intermediates—and they all produce qualitatively the same voltage dependence. Moreover, none of the ligands are required for voltage sensitivity, which persists when the ligands are removed altogether, and the channel is opened by crosslinking the PD to SecE or by PD deletion.

Rate changes of voltage driven channel closure that were observed upon PD immobilization suggested that sensor movements were either constricted due to the existence of a physical link between the sensor and the PD, or due a contribution of PD’s immobilized charged residue—the evolutionary conserved R74—to voltage sensing. When using site directed mutagenesis to add a charge to PD’s nearest neighbor, i.e. to the otherwise charge-free TM2b, we observed profound changes in the gating kinetics. The L80D mutant shows an accelerated gating suggesting that TM2b’s dipole may actually represent the voltage sensing element. Its helix dipole facilitates gating because its positive end points to the negative pole of the electric field (Figure 6). Yet, the wild-type exhibits slower gating kinetics because the dipole of its TM2b points in the opposite direction, i.e., opposes channel closure. This hypothesis is supported by the observation that the L80K mutation, which increases the dipole moment of wild type TM2b, decelerates gating. We conclude that gating is mediated by the dipole of a larger voltage sensor, of which helix2b is a part.

The indicated absolute values of the dipole moments (Figure 6) are not corrected for screening. Since the mutations localize to the middle of the membrane, i.e., since screening by exposure to the aqueous bulk is unlikely, we expect the general conclusions from the dipole calculations to be valid. Moreover, the screening corrections are known to be very important for longer helices [32], while TM2b spans only half of the membrane. 

TM2b is part of a larger hydrogen bonded network that glues helices 1–5 into a rigid body [33]. We speculate that it is this rigid body, i.e., the first half of the SecY protein may act as the voltage sensing element. This hypothesis is in line with (i) structural investigations that describe channel opening as a rigid body movement of SecY’s first half (helixes 1 to 5) relative to its second half [30] and (ii) the observation that nearly all of the conserved charged residues are located in the first half [33]. Exploiting *φ* to align the dipole of one or several helices constitutes a principal difference to voltage sensor movements in classical ion channels, where *φ* acts on a small number of charged residues. The benefit of targeting real charges is a much faster movement of the voltage sensor. Yet, fast channel gating is required for the transmission of action potentials or for sensing the ion channels, but it seems to be superfluous in the case of the protein translocation channel. It seems possible that the rather slow gating mechanism that relies on dipole alignment has evolved earlier in evolution than the fast gating mechanism of ion channels. This view is in line with a recent report about the gating mechanism of the β barrel protein lysenin [34]. As in the case of the protein translocation channel, the alignment of a molecular dipole is deemed responsible for gating.

Confirming the hypothesis that the dipole of TM1-5 acts as voltage sensor requires calculation of both the magnitude and the orientation of the collective dipole. This is possible only by molecular dynamics simulations, since screening effects [32] cannot be taken into account otherwise. In turn, it appears nontrivial to perform such simulations since high resolution structures of the channel in both its closed and open conformations are not available.

The eukaryotic translocon Sec61 requires accessory molecules like BiP [35] and calmodulin [36] to maintain the membrane barrier to small molecules during protein translocation. It resides in the endoplasmic reticulum, which is not known for permanently maintaining large *φ* values. In contrast, the bacterial translocon does not require accessory molecules. Membrane voltage is sufficient to seal the channel. Thus, Sec61′s voltage sensitivity [37,38] appears to be a mere relic of evolution. Voltage sensing in bacterial translocation may also be important to prevent backsliding of the polypeptide—a task that is carried out by BIP in eukaryotic protein translocation. Thus far, voltage sensitivity is missing in both the power stroke model [39] and the Brownian ratchet model [40] of protein translocation.

## 5. Conclusions

We have demonstrated that the voltage sensor of the bacterial protein translocation machinery is (i) an intrinsic part of SecYEG and (ii) includes the dipole of helix TM2b. The experiments required tight control over membrane voltage that was realized by investigating the purified and reconstituted translocation machinery in planar lipid bilayers. Voltage driven conformational transitions are key to distinguish between the power stroke and the Brownian ratchet mechanisms for protein membrane translocation in bacteria.

## Figures and Tables

**Figure 1 biomolecules-10-00078-f001:**
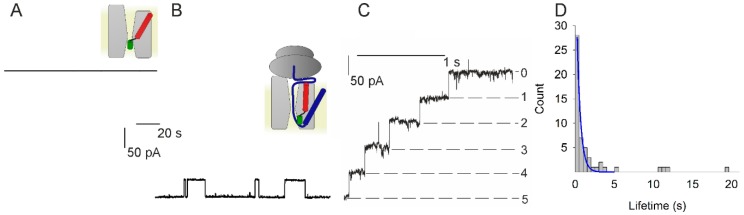
Channel activity of the SecYEG-RNC (ribosome-nascent chain) complex. (**A**) The planar bilayer did not show channel activity when only SecYEG-containing vesicles were added to the *cis* compartment (*φ* = −85 mV). (**B**) However, the subsequent addition of RNCs resulted in channel activity (*φ* = −85 mV). (**C**) Augmenting the membrane potential to *φ* = −140 mV resulted in channel closure. The negative potential induces the current to flow in the negative direction of the *y*-axis. Thus, the apparent movement upstairs indicates stepwise channel closure. The number of open channels is indicated on the right side. It decreases from five open channels at the moment of switching the potential to −140 mV, to zero open channels after little more than one second has elapsed. (**D**) Channel lifetime histogram for *φ* = −140 mV. From the single-exponential fit (blue) we extracted a channel lifetime of about 0.5 s. The colored schemes show SecYEG (light gray), the ribosome (dark grey), the nascent chain (blue), SecYEG’s helix 2b (red), and SecYEG’s plug domain (green).

**Figure 2 biomolecules-10-00078-f002:**
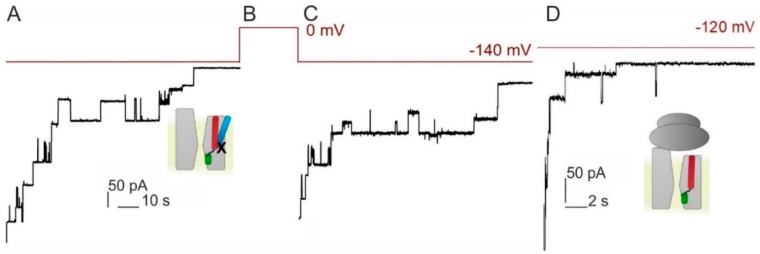
Voltage sensitivity is not conferred to the translocon by the signal peptide. (**A**–**C**) Closings of SecY(S329C)EG crosslinked to the signal peptide of OmpA (21C) at *φ* = −140 mV. The voltage protocol is shown in red above. (**D**) SecYEG in complex with empty ribosome also closes in response to voltage. Note the shorter time scale here as compared to (**A**,**C**). The small schemes obey the color code from Figure 1: with the signal peptide in light blue, nascent chain in blue and the cross-link depicted as X.

**Figure 3 biomolecules-10-00078-f003:**
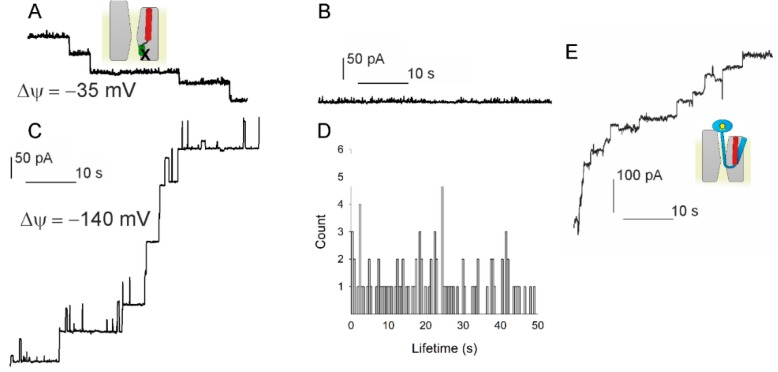
Testing the hypothesis that voltage driven plug movement confers voltage sensitivity to SecYEG (scheme in the upper left panel). (**A**) Crosslinking the plug to SecE by 1 mM potassium tetrathionate (KTT) in the SecY(F67C)E(S120C)G mutant forces reconstituted translocons to open. Single channels were recorded at *φ* = −35 mV. (**B**) At small *φ* values the channels virtually do not close. (**C**) *φ* = −140 mV elicits a conformational change that closes the channel. Scale bars for A–C are the same. (**D**) The channel lifetime histogram shows a wide scatter for −140 mV. (**E**) The plug deletion mutant SecYEG (Δ60–74) retains voltage sensitivity. Channel activity was observed when fusing the mutant containing proteoliposomes in the presence of SecA, ATP, Methotrexate (MTX), and proOmpA-DHFR. MTX binding to DHFR maintains the globular structure of the latter and thus prevents full translocation. *φ* = −85 mV closes the channels. The schematic representations use the same colour code as in Figure 1: proOmpA in light blue, MTX in complex with DHFR as a yellow pentagon inside a blue ellipse and the cross-link depicted as X. The experiments showed that plug immobilization or removal does not abolish SecYEG’s voltage sensitivity.

**Figure 4 biomolecules-10-00078-f004:**
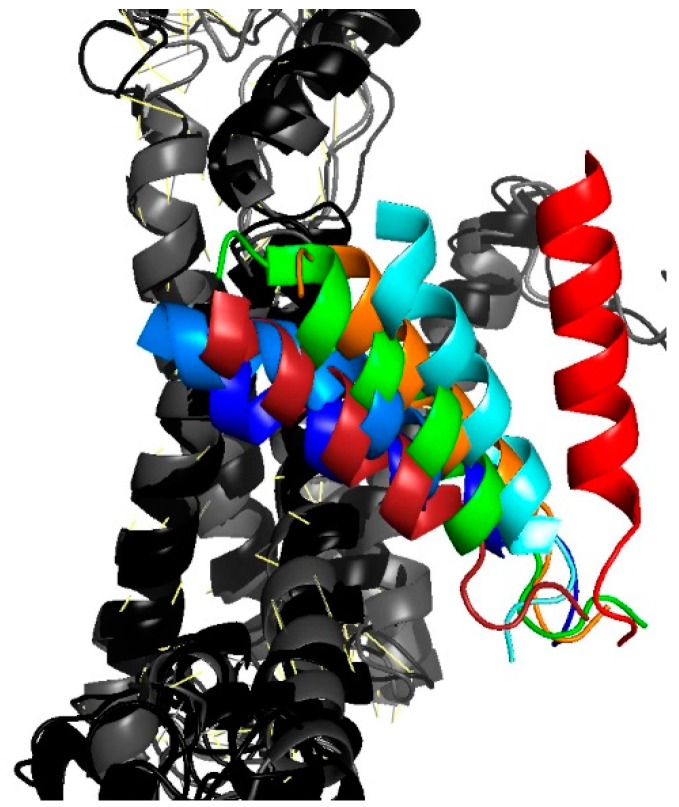
Movement of TM2b as gathered from an overlay of different translocon structures. The position of the colored TM2b is shown relative to the second half of the protein that consists of helixes 6 through 10 (in shades of gray). Closed conformations are represented by the PDB entries **4CG7** in dark blue and **4CG5** in light blue [9], **3J45** in dark red [28]. Partially or fully open conformations: **3J7R** in green [29] (Sec61-RNC), **5EUL** in orange [30] (SecYEG-pOA-SecA), **4CG6** in cyan [9] (RNC-SecYEG), **3J46** in red [28] (RNC-SecYEG).

**Figure 5 biomolecules-10-00078-f005:**
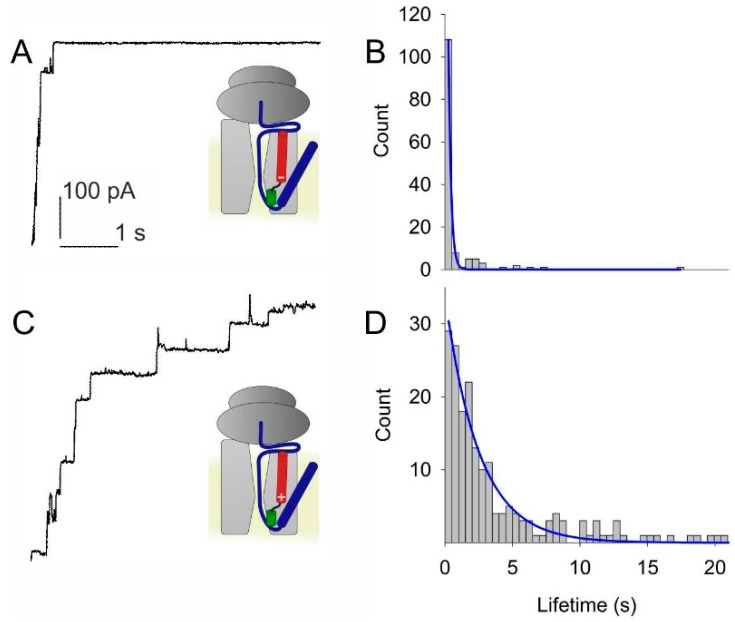
Test of the hypothesis that the partial charges at the poles of helix 2b act as voltage sensors. (**A**) Adding a negatively charged residue to the positive pole of the helix by site directed mutagenesis (L80D) accelerate voltage induced channel closings. (**B**) A single exponential fit (blue line) to the lifetime histogram indicates a channel lifetime of about 0.2 s at *φ* = −140 mV. (**C**) In contrast, closings at *φ* = −140 mV of SecYEG-RNC complex are slower with the addition of a positive residue to the tip of helix 2b (L80K). (**D**) The single exponential fit to the lifetime histogram shows a lifetime of about 2.5 s. The color code of the small schemes is the same as in Figure 1.

**Figure 6 biomolecules-10-00078-f006:**
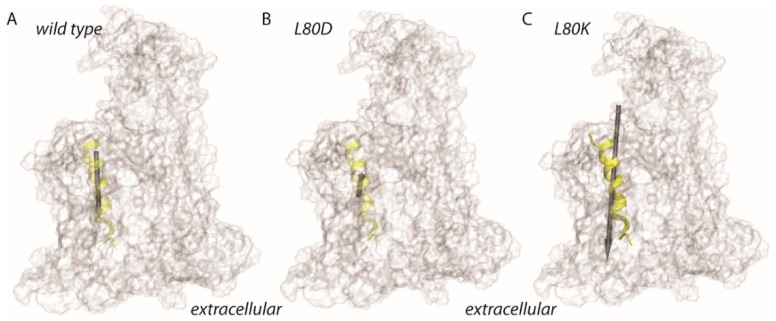
Schematic representation of the dipole moment estimates for TM2b of *E. coli* SecY. Dipole moments were estimated for the amino acid residues A79-P100, which are depicted as a transparent yellow ribbon. The protein is shown as a transparent surface. (**A**–**C**) Dipole moment estimates for helix 2 of wild type SecY (62.4D, panel **A**), L80D (25.7D, panel **B**), and for L80K (153.1D, panel **C**). The size of the gray arrow indicates the relative size of the dipole moments.

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
