# Peer review of "Voltage Sensing in Bacterial Protein Translocation"

_biomolecules, 2020, doi:10.3390/biom10010078_

Round 1

Reviewer 1 Report

Knyazev et al. have studied the voltage sensing ability of the translocon SecYEG. They find that despite lacking charged residues, TM2b (part of the lateral gate) is at least partially responsible for voltage sensing, likely through its dipole moment.

This is a solid paper, but I do have a few points that should be addressed.

I find this statement in the discussion regarding TMs 1-5 unsupported by the evidence:
"Our experiments suggest that it is this rigid body, i.e. the first half of the SecY protein may act as the voltage sensing element."
The experiments do point to TM2a and TM2b, but there is no reason to extrapolate to TM1-5. It's fine if the authors want to speculate, but that should be made clear. Absent more convincing evidence, the statement should also be removed from the abstract.

Which CHARMM force field was used? The citation is for CHARMM22, but that is now over 20 years old.

Minor points:
Line 17: "does not explain, why the..." no comma

Lines 20-21: "Yet, voltage driven closure..." this is not a sentence.

Line 26: "TM2b be part" should be "is part"

Line 147: "Close inspection of the only high-resolution crystal structure PDB ID:5GAE indicated that in the cryo-EM structure of SecY" This doesn't make any sense. It's not a crystal structure; also, the phrase "in the cryo-EM structure of SecY" is unnecessary in this sentence.

Line 230: Starting a new section with "This result..." is confusing. What is "this" referring to?

Figure 4 caption: "Partially of fully" should be or

Line 290-292: This is not a sentence.

Reviewer 2 Report

The SecYEG translocon mediates the translocation of secretory proteins and the integration of membrane proteins across and into the bacterial inner membrane, respectively. The polytopic, channel-forming SecY subunit forms an hourglass-shaped pore that is constricted at the center by a hydrophobic ring (HR) of conserved nonpolar residues, with a short helix (TM2b) connected to a plug domain (PD). The resting channel is sealed by the PD, and deletion of this helix renders the translocon permeable to small molecules. The channel is activated by ligands, either ribosomes (co-translational mode) or signal sequences (post-translational mode), to displace the PD and gate open the channel. The HR serves as a gasket-like seal around polypeptides in transit, thereby maintaining the membrane permeability barrier to small molecules. Although the structural features of the SecYEG channel have been extensively investigated by several groups for years, the mechanism of voltage sensing within this complex has remained enigmatic. Previous work has shown that ligands must be present to effect voltage-dependent SecYEG channel activity, suggesting that voltage sensitivity may originate from ligands. Whether other voltage-sensing structural elements may be present within the SecYEG channel itself remains unknown.

The goal of the present study is to identify region(s) of the SecYEG complex that may serve to sense transmembrane voltage and gate the channel. To this end, Knyazev et al. perform electrophysiological measurements of purified and reconstituted SecYEG complexes (wild-type and mutant) in the presence and absence of ligands. The central conclusions of this work are that SecYEG contains an intrinsic voltage sensor, that TM2b constitutes part of the voltage sensing mechanism, and that alignment of the helix dipole in the transmembrane electric field underpins the voltage sensing mechanism. The potential significance of this work arises from the attainment of new and important structure-function information on a protein translocase that is conserved in all kingdoms of life. However, while the study addresses a fundamental mechanism, and the electrophysiological approaches appear to be sound, there are significant technical and conceptual weaknesses, listed below. Most notable among these weak points is the lack of experimental evidence to support the role of TM2b as a voltage sensor and ambiguity as to how a dipole-based sensor mechanism would result in identical gating behavior regardless of the direction of the applied electric field.

Major points

1) The voltage-sensitive closure of the reconstituted translocon is shown for a number of bound states (SecYEG-RNC, SecYEG-ribosome, SecYEG-preprotein) but channel lifetime histograms are shown for only one such state (Fig. 1D). Lifetime histograms (and single exponential fits) should be shown for all translocon states described in this study.

2) A major underlying premise of this manuscript is that the gating behavior of the reconstituted SecYEG channel is the same regardless of the orientation of the applied electric field. This assumption (lines 201-203) is based on a comparison of this study with previously published work with planar bilayer systems. For this analysis to be as rigorous as possible, the authors should repeat some of the channel closure experiments with the applied electric field in the opposite orientation, to confirm that this holds true for their system.

3) With respect to point 2, one major assumption of this work (lines 203-306) is that because channel closure occurs identically regardless of the field orientation, the voltage sensor may consist of an electric dipole on a structural element rather than discrete point charges. However, this reasoning must be explained more thoroughly. The orientation of a fixed dipole within an electric field would be expected to have just as much dependence on the direction of the field lines as point charges would. It is not clear why a dipole-based voltage sensor would effect identical gating behavior regardless of field orientation.

4) This study would be strengthened by inclusion of control experiments. For instance, for the crosslinking-based work in Figure 2, current traces of the SecY(S329C)EG construct should be shown in the absence of crosslinked OmpA(21C) for direct comparison of gating behavior in the presence and absence of covalently bound signal peptide (i.e., to show that this SecYEG mutant displays wild type gating without crosslinked substrate).

5) If the helical macrodipole of TM2b (which lacks ionizable side chains) is a central part of the voltage sensor, then the helix should have partial positive charge density toward the N-terminal pole and partial negative charge density toward the C-terminus of the helix. The substitution of the N-terminal site L80 with either K or D does indeed alter gating kinetics, supporting a role for these charged sited in channel gating. But given that L80D would likely diminish the charge separation in a TM2b macrodipole, and that L80K would enhance the charge separation (as depicted in Figure 6), do the authors have an explanation for why the L80D construct shows accelerated closure kinetics whereas the L80K construct shows decelerated kinetics, particularly since the orientation of the field does not appear to matter? Can these results be just as readily explained by mutation-based disruption (or reinforcement) of an ionic network that stabilizes the channel, with an actual voltage sensor positioned elsewhere? Furthermore, if voltage sensitivity is independent of the direction of the applied electric field, would the authors expect to obtain similar results if the substitution mutations were introduced at the opposing end of the helix? If the applied electric field were refersed? This would help substantiate their proposed mechanism.

6) The authors cite as their secondary conclusion (lines 301-302) that the voltage sensor is comprised of a rigid body assembly of TMS1-5. But there no experiments in the manuscript that test this – this idea arises from speculation in the Discussion. By the authors own description, evaluation of such a collective dipole would require computational approaches beyond the scope of this study (lines 285-289).

Minor points

1) Supplementary figures S1 and S2 are incorrectly matched with their corresponding references in the main text. In the main text, there is no reference to the figure shown in Figure S1.

2) The traces in Figures 3A and 3C should have current (y-axis) and time (x-axis) scales.

Reviewer 3 Report

Paper looks fine to me. Would be improved with a bit more editing and clarifying

eg - spheroplast (not spheroblast)

And in abstract "However, this hypothesis....." to replace "However, hypothesis....."

My main problem was in digesting the data for a non-specialist. Perhaps a bit more description/ schematic assistance in the figure would help a non-expert of membrane conductance. For example, from the very outset it is not clear (at least to this reviewer) why Fig 1C demonstrates channel closure.

Round 2

Reviewer 2 Report

The authors have addressed the concerns raised. This manuscript is suitable for publication in Biomolecules.